# Composite Reinforcement Architectures: A Review of Field-Assisted Additive Manufacturing for Polymers

**Madhuparna Roy [1,2], Phong Tran [1,2], Tarik Dickens [1,2,\*] and Amanda Schrand [3]**

[1] High-Performance Materials Institute, 2005 Levy Avenue, Tallahassee, FL 32310, USA; mr13@my.fsu.edu (M.R.); phong.tran@eng.famu.fsu.edu (P.T.)

[2] Department of Industrial & Manufacturing Engineering, FAMU-FSU College of Engineering, 2525 Pottsdamer Street, Tallahassee, FL 32310, USA

[3] Air Force Research Laboratory Munitions Directorate, Fuzes Branch, 306 W. Eglin Blvd, Eglin Air Force Base, FL 32542-6810, USA; amanda.schrand.2@us.af.mil

\* Correspondence: dickens@eng.famu.fsu.edu

**Abstract:** The demand for additively manufactured polymer composites with increased specific properties and functional microstructure has drastically increased over the past decade. The ability to manufacture complex designs that can maximize strength while reducing weight in an automated fashion has made 3D-printed composites a popular research target in the field of engineering. However, a significant amount of understanding and basic research is still necessary to decode the fundamental process mechanisms of combining enhanced functionality and additively manufactured composites. In this review, external field-assisted additive manufacturing techniques for polymer composites are discussed with respect to (1) self-assembly into complex microstructures, (2) control of fiber orientation for improved interlayer mechanical properties, and (3) incorporation of multi-functionalities such as electrical conductivity, self-healing, sensing, and other functional capabilities. A comparison between reinforcement shapes and the type of external field used to achieve mechanical property improvements in printed composites is addressed. Research has shown the use of such materials in the production of parts exhibiting high strength-to-weight ratio for use in aerospace and automotive fields, sensors for monitoring stress and conducting electricity, and the production of flexible batteries.

**Keywords:** additive manufacturing; field-assisted; tunable properties; fiber orientation control; multifunctional composites; printed composites; microstructure

## 1. Introduction

In recent years, additive manufacturing has seen a rise in the development of low-priced extrusion-based printers, particularly for enthusiasts and designers in small office settings [1,2]. These printers provide an affordable platform to create prototype models quickly and economically, leading to an overall growth in the 3D printing industry and have been used for applications in the field of medicine [3,4], development of various military and civilian applications [5–7], microelectronics and sensing systems [8–10], and in education [11–13]. Extrusion printers today are expanding their material database from basic thermoplastic polymers (Acrylonitrile Butadiene Styrene (ABS), Polystyrene (PS), Polylactic Acid (PLA), etc.) to high-temperature polymers (Polyether Ether Ketone (PEEK), Polyetherimide (PEI)), thermoset materials, semi-crystalline [14,15] as well as metals [16–19], ceramic [20–22], and composite materials [23–25]. Although this manufacturing process allows customization of macro features (i.e., in the millimeter to centimeter range), it lacks the ability to tailor

microscopic features (i.e., in sub-millimeter or micrometer and nanometer range) within the material which can ultimately be used to enhance functional properties such as strength, in localized areas.

The concept of building structures from the bottom-up using direct writing (DW) has been around since chemical vapor deposition (CVD) and physical vapor deposition (PVD) were introduced [26,27]. Although these processes allow for structuring and patterning, they require high temperatures, high power, a certain level of purity and several hours, which is seldom suitable for all types of substrates and materials. CVD has also shown to produce tailored shapes and structures such as 2D triangles, 3D pyramids, and hexagons just by varying the vertical distance between the substrate and the precursor, which is also a major determining factor for the quality and microstructure of an additively manufactured part [28,29]. Techniques such as aerosol jet process, and precision syringe-based nozzle dispensing processes, have the potential to be used for the production of high-resolution 3D microstructures [30]. However, scalability is a hindering factor when it comes to these processes as they are typically perceived as 2D/2.5D manufacturing. Aerosol jet has the capability to produce prints with 2-4X higher accuracy than inkjet, in the range of 10 microns, while being able to print on non-flat and non-smooth surfaces, which is typically unachievable for inkjet [31]. In contrast, processes such as stereolithography (SLA), selective laser sintering (SLS), 3D printing (3DP), fused filament fabrication (FFF) and laminated object manufacturing (LOM) have been classified as a group of technologies that have the potential to efficiently fabricate intricate and complex 3D microstructures.

## 1.1. How Composites Are Finding Their Way into Traditional AM

Composite microstructure designs draw inspiration from exotic structures and materials found in nature and have shown to reduce stress concentrations near discontinuities because of the heterogeneous arrangement of reinforcement. An example of such a structure is balsa wood, which is a lightweight naturally occurring composite with specific bending stiffness and specific bending strength comparable to those of engineering materials [32]. Cellulose fibers and a lignin-hemicellulose matrix in cellular structures are the primary constituents of wood and bamboo, thus proving that chemical composition, nano/microstructure, and architecture are the key elements in designing novel materials with tunable properties [33]. Studying the fundamental mechanism behind these structures has led to an interest in using 3D printing to replicate these strong yet lightweight structures [30,33–38].

Materials used in AM have evolved over the last few decades, with the addition of reinforcement to the existing polymers, and are tailored for the fabrication of advanced, multifunctional parts with improved mechanical properties [39–42], thermal conductivity [43–46], reduced coefficient of thermal expansion [43,47], improved dielectric permittivity and controllable resonance frequency [48,49] using additive manufacturing. In plastic-based composites, fibers have been successfully used to reinforce the plastic for over 50 years to achieve enhanced mechanical properties [50]. The use of composite materials in AM was expected to enhance the strength of the end-use parts, which is true in some cases depending on the relationship between the load-bearing axis and the axis of build. The combination of high-modulus and high strength fibers with a polymer matrix produces a composite with high stiffness, strength, and lower coefficient of thermal expansion, nevertheless previous attempts to randomly introduce composites to additive manufacturing methods have raised many limitations and issues with regards to fiber size, non-uniformity, the ability of the matrix material to hold its shape after deposition, and energy absorption of metal fibers, to name a few [37,51–53]. Several companies such as Fortify, Markforged, Cosine Additive, and others have introduced 3D printers into the market with the capability to manufacture composite parts to enhance the mechanical properties of the polymer typically used [54–56]. Most extrusion printers using thermoplastic filaments reinforced with chopped glass or carbon fiber produce parts with these fibers aligned in the direction of print due to shear forces experienced inside the nozzle [57]. Therefore, the ability to truly control the microstructure of the materials in these processes is critical to the production of parts with customizable and localized properties. The use of a manufacturing process with an external source of energy, such as an externally applied field to control the microstructure could enable the same material to

provide different functions such as flexible/rigid, conductive/non-conductive properties and potentially eliminate the need for using two or more materials for building the same component, while imparting improved mechanical properties.

The use of external fields to control the directionality or orientation of particles or fibers introduced into polymeric matrices have been sparingly investigated over the past decade. Materials with such capabilities have shown immense promise in the aspect of imparting properties such as self-healing, self-assembly and improved strength in localized areas of a composite part in a non-invasive manner during the manufacturing process [53,57–62]. In addition, such composites have also seen applications in electronics for their controllable conductivity, capacitance and optical properties with an advantage of roll-to-roll on-demand production capabilities [63–65]. For additively manufactured composites to possess the same level of mechanical strength as bulk material while having multifunctional properties, a thorough understanding of interfacial bonds between consecutive printed layers is necessary. This paper presents a comprehensive review of the state-of-the-art micro-additive manufacturing processes that effectively use external sources of energy or fields to achieve complex internal structures and multi-functionality in polymer composite structures, primarily focused from a materials point of view and their compatibility with specific field-assisted additive manufacturing techniques.

Although the ASTM 52900 standard for additive manufacturing categorizes the technology into seven families based on process and material, this paper is focused on processes involving polymer composites viz. vat photopolymerization or SLA, DW and extrusion [66]. The different processes yield a wide range of print resolution, which in turn can affect the mechanical and functional integrity of the final part. Figure 1 breaks down additive manufacturing techniques based on their material, process, and structure to ultimately relate down to its properties. Field-assisted manufacturing processes are confined to DW, material jetting, and extrusion of composites and the fillers are manipulated either by magnetic, electric, or acoustic fields. The external fields are used to alter the microstructure of the printed part which relies on the shape, alignment, and concentration of the filler in the matrix of the composite. The macrostructure defines the interlayer adhesion, porosity, and the processing temperatures, all of which contribute towards the properties of the final print. The figure portrays the relationship between each block and how properties are a result of a combination of materials, process and micro/macrostructure.

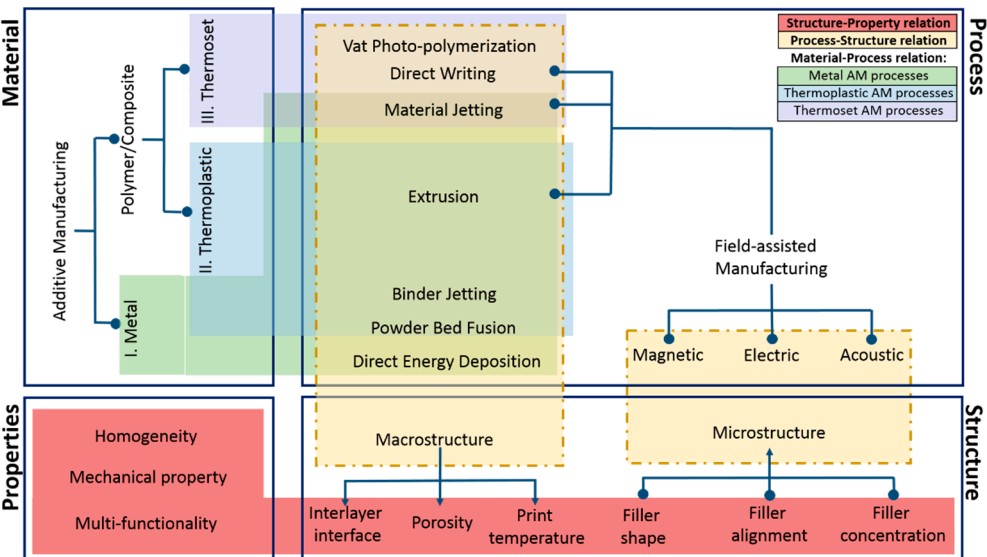

**Figure 1.** Material-process-structure-property relationship classifying AM processes.

*1.2. Need for Fiber Orientation Control in Composite AM*

The need for fiber orientation control is crucial to produce high-performance composite structures. This is because the aligned fibers allow a point-load to be dispersed along the length of the fibers oriented in the same direction so that the loads are transferred away from critical locations [67]. When a composite is formed, the properties of the resultant structure have properties based on the loading and orientation of the constituent materials. Since AM allows for selective material deposition, the ability to control fiber placement can further enhance the properties of the printed part.

Traditional polymer matrix composites are predominantly reinforced with ceramic, metal or polymeric one-dimensional (1D) reinforcement such as glass, steel, aramid (Kevlar) or continuous carbon fiber, typically 10 microns in diameter and provide strength and stiffness along the long axis of the reinforcement. However, these 1D reinforcements impart no strength in the other two axes and can be overcome by layering 1D fibers at different angles on the plane, performed in traditional composite manufacturing techniques such as hand lay-up and vacuum bagging. Continuous fibers are woven into 2D arrays which can be layered to provide strength in two axes, which still leaves the possibility of fiber layer delamination due to lack of reinforcement in the third axis. 3D reinforcement solutions published include the insertion of out-of-plane fibers by mechanical punching [68–70], weaving, braiding, knitting [71] and the growth of aligned CNT on the surface of 2D woven fibers, often referred to as z-pinning [72–74]. Hence, the property of the composite is not only affected by the individual material constituents of the composite but rather the interfacial properties of the materials which further defines the effectiveness of stress transfer and determines other factors such as impact toughness, off-axis strength, and overall functional performance. Z-pinning in AM has shown to reduce mechanical anisotropy and to improve strength between layers [75–79]. Essentium's new CNT-coated filament, which uses microwave induction heating to merge consecutive layers after a part is printed, Garcia et al. implemented aligned CNT forests that can bridge and strengthen this interlaminar region and Wicks et al. used the concept of the so-called "fuzzy fiber" (CNTs grown on carbon fibers) and applied it to composite laminates can provide both interlaminar and intralaminar reinforcement as seen in Figure 2 [67,80,81]. Design for Additive Manufacturing (DFAM) is another aspect of this manufacturing process that focuses on exploring design concepts that are not considered for traditional Design for Manufacturing (DFM). DFAM draws a relationship between processes, structure, and property to ultimately determine the behavior of the final part by addressing issues at the microstructure level [82].

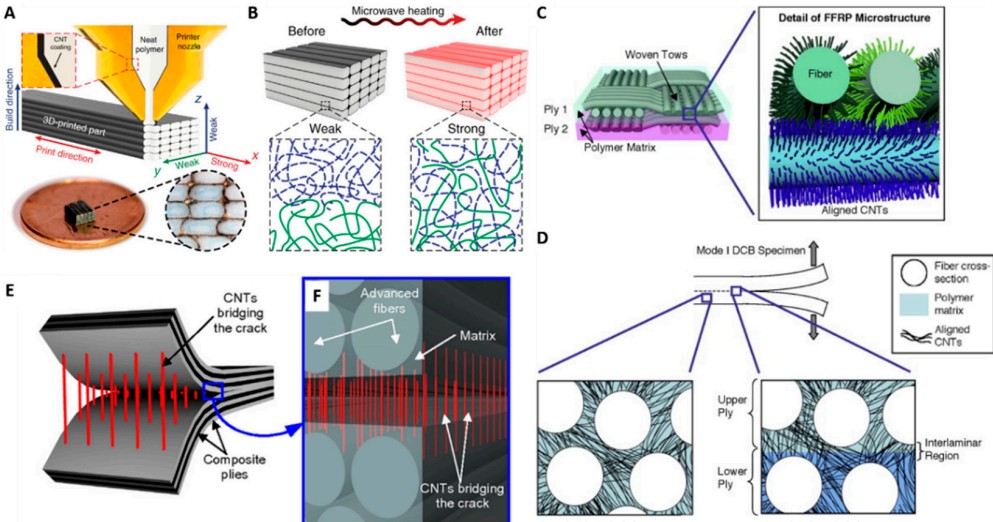

**Figure 2.** (**A**,**B**) Essentium's process of filling voids formed during extrusion with the use of CNT-coated thermoplastic filament [83], (**C**) Radially aligned CNTs grown on advanced fiber cloth, (**D**) CNT intralaminar and interlaminar reinforcement [80], (**E**,**F**) use of aligned CNT forests to strengthen interlaminar region in composite laminates [67].

In this review, we assess a non-traditional approach to additive manufacturing termed as "field-assisted additive manufacturing". The ability to program complex architectures of micro and nanoparticles of different shapes and sizes to exhibit high strength in composites makes this manufacturing process promising to produce tunable properties in materials.

## 2. Field-Assisted Manufacturing Technologies

Field-assisted additive manufacturing is a technique that has been adopted into a few AM processes that primarily make use of polymer composite materials. It can be classified into three major groups, including electric, magnetic, and acoustic fields. The ultimate goal of field-assisted additive manufacturing is to separate and control individual fiber's placement and orientation in a polymer matrix to fine-tune properties of the final part, thus creating the possibility of self-assembling, organized structures, where they behave as building blocks. This variation of additive manufacturing demonstrates the feasibility of controlling fiber orientation either during the process of deposition or after the material is deposited and before it is cured to retain the microstructure [53,57–59,84–87]. Rheological properties of the matrix also determine the ability to control fiber orientation and have been used in the range from 0.9 to over 3 Pa.s. Formulated inks are required to have distinct flow responses that range from pure Newtonian for orientation control to viscoelastic for shape retention and control. Another important factor while realigning particles using an external field is determining the amount of time it takes to reorient a particle depending on the field strength and the viscosity of the matrix material. The benefits of alignment range from an increased tensile response and increased conductivity to heat dissipation for consideration as material replacements for metal parts [85]. Research has shown cylindrical/rod/fiber-shaped particles to be more effective in improving mechanical strengths of composites compares to spherical, disc, and toroid shaped particles. A greater strengthening effect is possible for cylindrical reinforcement of higher aspect ratio at higher rates and at high volume fractions of up to 60% [88].

Field-assisted manufacturing was originally introduced as a non-AM process involving molds and external fields as seen in examples in Figure 3. As expected, the inclusion of fibers with controlled alignment imparts enhanced mechanical properties in addition to being able to provide localized patterns. Field-assisted AM adopted from molding techniques have several advantages including the ability to produce functionally graded structures and optimized microstructures for specific functions and applications, such as in 4D printing-a manufacturing process involving multiple materials, and structures capable of transforming from one shape to another is known for using external fields or stimuli after the completion of print to induce shape changes, thus allowing custom reinforcement architecture in each layer [89–91]. Fine-tuning the position of reinforcement has led to the production of actuators and soft devices that can undergo programmed shape changes when triggered with an external stimulus (expansion and/or contraction) [53], self-healing capabilities in composites and electrical circuitry [35,92] improved conductivity in transparent thin films [93] and self-assembly of nanostructures [85,93–96]. This has made AM a manufacturing process that allows the integration of controlled design synthesis methods to adjust corresponding manufacturability outcomes. The use and effects of each type of externally applied fields on composites materials used in AM are discussed in the following sections.

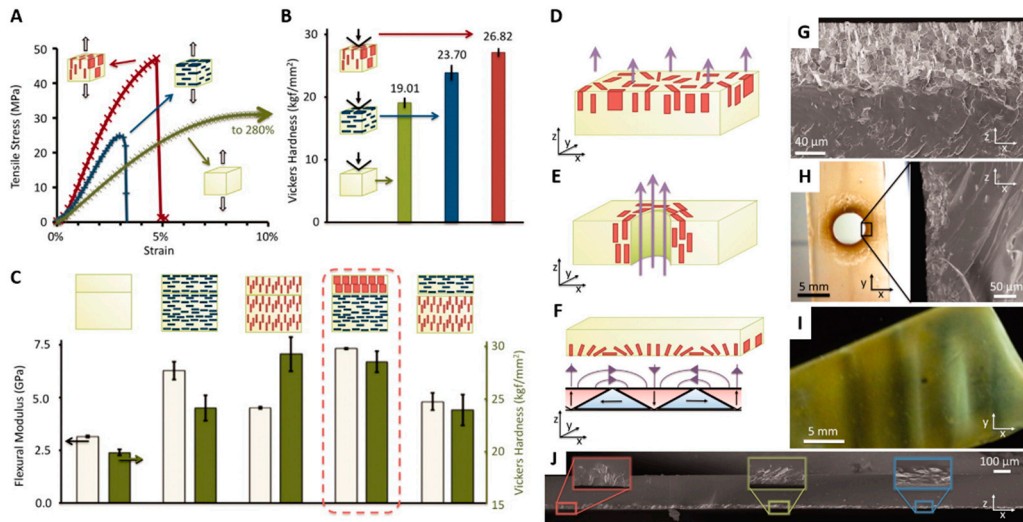

**Figure 3.** (**A**) Improvement in tensile strength for polymer with 20 vol.% Al$_2$O$_3$ platelets parallel with applied force (red) vs. perpendicular (blue) and non-reinforced (green) samples (**B**) Increased hardness for reinforced resins with 10 vol.% Al$_2$O$_3$ platelets parallel to the applied load versus perpendicular and non-reinforced samples (**C**) Flexural modulus and out-of-plane hardness of bilayer rectangular samples with different combinations of reinforcement orientation in samples produced using field-assisted, non-AM methods (**D**) Particles aligned along z-axis due to applied field (**G**) SEM of particles aligned in z-axis (**E**) Reinforcing particles concentrated around a hole (**H**) SEM of particles distributed around hole in part (**F**) Graduated particle alignment achieved by refrigerator magnets (**I**) Topological ripples formed in a polymer substrate due to particle alignment (**J**) SEM image of Mohawk-like structure in layer [74].

## 2.1. Magnetic Field-Assisted Additive Manufacturing

When a magnetic field is applied during the process of building each layer in an additively manufactured part, it can align magnetic reinforcement along the field lines. Magnetic reinforcements are typically in the nanometer to micron range, and can be entirely made of a magnetic material, or can have magnetic nanoparticle clusters coated onto a non-magnetic material, thus giving them the ability to form complex microstructures when a field is applied. While printing with composites, the viscosity of the matrix material plays a major role as it determines whether or not the reinforcement will remain suspended in the matrix when left undisturbed. The viscosity also determines the required magnetic field to achieve the rotational control of a particle during flow. It is understood that the torque produced due to the applied field needs to be greater than the shear forces experienced by the particle. The aspect ratio of the reinforcement also plays a major role in the torque experienced. Using a torque balancing equation, the magnetic, gravitational, viscous torques acting on magnetic reinforcement can be determined [35].

Stereolithography is widely used in conjunction with an externally applied magnetic field since the fiber reorientation takes place in a vat of liquid photopolymer, which is cured, layer-by-layer with the help of a UV source or laser beam as the part is submerged. Nakamoto et al. incorporated 0.5 vol.% ferromagnetic short fibers in a photopolymer matrix in a SLA system equipped with a magnetic field setup with four horizontal and one vertical electromagnet to vary the magnitude and direction of the magnetic field applied [58]. A 15 mT total magnetic flux applied during the experiment proved to be much lower than the desired field strength to reorient 0.32 μm long and 0.04 μm diameter fibers in a 0.36 Pa.s viscosity matrix. The fibers were able to orient themselves vertically in the applied 15 mT only after they were pre-magnetized in a 150 mT field. Nevertheless, the results showed fiber reinforcement in the Z-direction in the printed substrate, similar to z-pinning in traditional composite manufacturing, although some issues recorded with the process included incomplete sure of the photopolymer due to shadows cast by the short fibers within, also resulting in fiber pullout.

Similar research conducted by Martin et al. [59] implemented a magnetic field to align rod-shaped particles radially around a circular defect and at 0, 30, 60, and 90 degrees orientations, to study the effects. Tuning the placement of reinforcement based on part geometry and load direction has been proved to be more effective than unidirectionally reinforced composites due to the load distribution. Since the "osteon-inspired" radial architecture is symmetric, the load can be applied at any angle relative to the microstructure to obtain similarly high performance. Tensile samples printed with the aforementioned fiber orientations were compared using FEA techniques and experimentally validated to show radial orientation around a circular defect to provide maximum strength, shown in Figure 4. The strength of the samples is seen to gradually increase as the alignment of the fibers transition from 90 to 0 degrees, or in other words, the fibers are aligned in the direction of the applied load. Lu et al. implemented a combination of SLA and extrusion to selectively place and align magnetic particles in their parts thus enabling precisely controlled particle filling patterns and ratios to achieve heterogeneous properties [97].

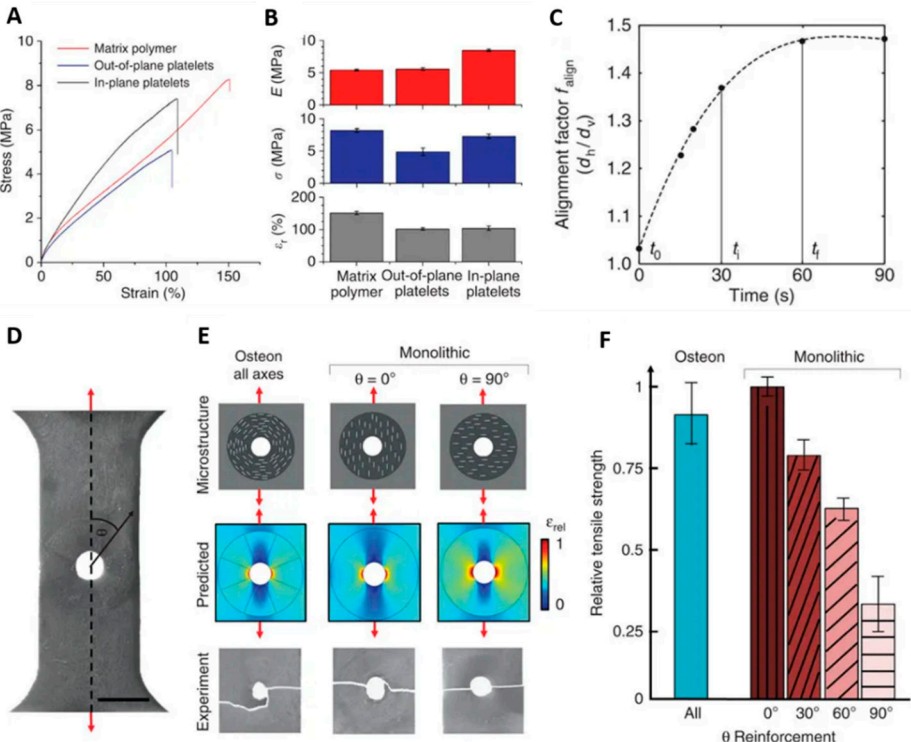

**Figure 4.** (**A**) Stress-strain curves of magnetically aligned platelets in different orientations (**B**) elastic modulus (**E**), strength ($\sigma$) and strain-at-rupture ($\varepsilon_r$) of printed samples (**C**) alignment dynamics of platelets in the presence of a rotating magnetic field and the increase in the degree of platelet alignment as a function of time [53] (**D**) Samples with circular defects are 3D magnetic printed with programmable reinforcement architectures including "osteon-inspired" microstructures with concentric reinforcement orientation and "monolithic" microstructures as shown in (**E**). (**E**) "Osteon-inspired" microstructures are predicted to exhibit less relative strain compared with misaligned "monolithic" microstructures. (**F**) Tensile tests of printed composites with circular defects show that "osteon-inspired" architectures out-perform all but the perfectly aligned "monolithic" sample. As the "osteon-inspired" architecture is symmetric, the load can be applied at any angle relative to the microstructure to obtain similar performance. Scale bar, 5 mm in (**D**) [59].

The concept of using an external field-assisted printing scheme was simulated by Martin et al. to study the fundamental mechanism implemented for extrusion printing in an applied magnetic field. It involves overcoming the shear forces of the matrix developed in the flow channel so that the applied field can take effect. The viscosity of the matrix affects the aspect of quickly reorienting

the reinforcement in-flow as well as holding the position of the reinforcement fibers after material deposition and prior to cure. The apparatus consisted of a flow channel and electromagnets with fields ranging from 40 mT to 80 mT on either side. These were used on a composite containing 50 μm length and 2 μm diameter fibers in solvents (glycerol and Isobornyl Acrylate) of viscosities 1.41 Pa.s and 0.0075 Pa.s respectively [59]. Although the goal of the project was to arrest Jeffrey orbits by applying magnetic fields to prevent fibers from continuously rotating during extrusion resulting in a randomized fiber architecture, in-flow fiber alignment was achieved. The strength of the printed composite with aligned fiber, however, could not be verified due to the absence of any mechanical tests for the experiment.

Kokkinis et al. [53] reported inks preloaded with stiff magnetized pellets and a magnetic field produced by neodymium magnets were used to achieve alignment. Using direct ink writing technology and four syringes containing a different composition of magnetic material to produce inks with different formulations, multi-material prints are attainable. The apparatus included a two-component mixing and dispensing unit for gradual change in ink composition. This allowed for control over-concentration of filler material in different parts of a structure. The base resin of 0.2 Pa.s viscosity Newtonian fluid made it convenient to reorient magnetic particles in magnetic fields of 1–10 mT. As the concentration of particles was raised to 2–8 vol.%, the fluid changed to show viscoelastic properties with increased yield stress and low shear-viscosity. This required an increased magnetic field of 40 mT and 60 s for the particles to achieve their final orientation. The relationship between the viscosity of the resin and the applied magnetic field is critical; the higher the viscosity of the fluid, the stronger the magnetic field needs to be in to influence the magnetic particles. The effect of aligned reinforcement is seen in Figure 4.

A slightly different extrusion manufacturing technique for magnetic fluids is Drop-On-Demand or DOD. DOD is primarily used to print droplets in gaseous environments. However, Vekselman et al. made use of this technology to print droplets with magnetic particles. Printing magnetic fluids on demand by applying a magnetic field is difficult because the fluid tends to form a long liquid bridge between the nozzle and the drop—which can be overcome by printing droplets inside another gaseous or liquid media. Their study mostly discussed the process of separating the magnetic droplet from the nozzle using an external magnetic field and no mechanical properties were tested [95]. Isaza et al. also used DOD with a different approach with the assistance of a coil gun to print magnetic inks inside gaseous or liquid media, while possessing the ability to control droplet size, and hence, the resolution of the print [94].

Magnetic field-assisted composites have also been used for imparting self-healing properties to materials. Magnetic microparticles ($Nd_2Fe_{14}B$), which have permanent magnetic properties as opposed to iron oxide particles, which require an external field, were used in ink, in combination with Carbon Black for improved conductivity. The magnetic particles used were approximately 5 μm in size and were obtained by grinding commercially available magnets. The ink was printed in the presence of an external magnetic field to orient the particles along the direction of the trace/print line, with the resulting printed line behaving like a permanent magnet [92]. After printing, the bar magnet and platform were left untouched and unmoved for 15 min to allow the magnetic particles to orient along the direction of the external magnetic field. Perhaps the use of a lower viscosity matrix and a stronger magnetic field would have expedited the alignment process. Huber et al. explored a similar avenue with the inclusion of $Nd_2Fe_{14}B$ powder in a thermoplastic matrix to print polymer-bonded magnetic systems with complex shapes and locally tailored magnetic properties. The accuracy and performance of the magnet were validated by comparing it to a finite element model and comparing hysteresis curves with injection-molded samples. It was concluded that due to the voids present in a printed structure, the lower volumetric mass resulted in a lower magnetic remanence [98].

Kim et al. [99] were able to achieve a more complex design of 2D planar and 3D structures with programmable ferromagnetic domains using a viscoelastic ink that included fumed silica particles as a rheological modifier. Their designs show advanced applications in soft electronic devices, structures capable of "stopping" and "holding" a fast-moving object, and the ability to wrap and transport a



small object such as a pharmaceutical pill using rolling locomotion under a rotating magnetic field. The process of printing such structures also involved extrusion of an ink with neodymium-iron-boron particles through conical micro-nozzles with a 50 mT magnetic field applied at the point of exit using an electromagnet or a permanent magnet. Printed magnetic actuators with carbonyl iron beads have also shown the potential to behave as soft intelligent structures that can be programmed to reshape and reconfigure when exposed to a magnetic field and are capable of floating on water [100]. However, such materials [100,101] are primarily used for shape-memory applications and can be categorized as 4D printing rather than improved mechanical properties in AM. Although magnetic particles are typically used for prints in the millimeter to micrometer range, the Oakridge National Lab has implemented a composite material with 65 vol.% of NdFeB in Nylon-12 to manufacture large magnets using their Big Area Additive Manufacturing (BAAM) system and compared results with traditional injection-molded magnets to prove that BAAM is capable of producing a better hysteresis loop and higher intrinsic coercivity and remanence values [102]. Although these parts are manufacturing using AM, they are not classified as field-assisted manufacturing due to the fact that no external source was used to control the position or alignment of the particles during manufacturing [89,96,98,102–107]. Many other articles related to magnetic particle or fiber alignment refer to a process involving molds in to which the composite material is cast and is exposed to magnetic fields for long durations of time to achieve the desired alignment in the part- and hence avoid all the challenges introduced by AM [74,84,108,109].

Among the broad range of nanomaterials that are currently being used in biomedical applications, 1D nanoparticles are of special interest due to their high aspect ratio, ease of magnetic manipulation at low magnetic fields and ability to impart higher mechanical properties [110]. Nickel nanowires have also been reported to possess strong magnetic moments as well as magnetic anisotropy compared to their spherical counterparts [111,112]. A potential application of magnetic nanowires in the medical field is focused towards magnetic tweezers for cell manipulation under low magnetic fields [113,114]. Also, with the recent growth of smart electronic devices that rely on wireless communications such as smartphones and watches, the demand for smaller and faster integrated sensors and antennae have also increased. The integration of on-chip inductors and antennae with magnetic materials of high permeability offers potential benefits to reduce the size of a device [88].

In summary, only a limited number of articles have been published in the realm of additive manufacturing of composites with the assistance of an external magnetic field to control fiber placement, of which an even smaller number report on the mechanical properties of the produced composite with respect to fiber orientation, which is of immense interest.

## 2.2. Electric Field Assisted Additive Manufacturing

Electric field-assisted manufacturing can be described as a process wherein the position and microstructure of reinforcement in a composite are controlled by the application of an electric field; meaning that the reinforcement must be receptive to such an applied field. It follows the same rules as magnetic field-assisted manufacturing, except the reinforcing element and the applied field vary. The applied field determines the torque, which is the driving force that rotates the particle to line up with the direction of the electric field. Kim et al. and Holmes et al. used the following equations to estimate the amount of time it would take to rotate a particle as well as to form chains in the fluid matrix based on the field strength and viscosity when using spherical particles:

$$t_{rotate} \approx \frac{10^2 \eta}{\varepsilon_0 E^2} \tag{1}$$

$$t_{chain} \approx \frac{10 \eta}{\varepsilon_0 E^2} \tag{2}$$

where *t* is in seconds, viscosity, $\eta$ is in Pa.s. and electric field is in *V/m* [110,115]. Lower rotation times are ideally preferred to increase the overall speed of the manufacturing process.

Similar to the orientation of fibers in Martin's research in Section 2.1, a Field-Aided Micro-Tailoring (FAiMTa) involves curing a photopolymer containing randomly dispersed micro- or nano-sized particles, fibers or platelets in the presence of an electric field, instead of a magnetic field. The goal of this research was to reduce stress concentrations by aligning fillers based on the geometry of the composite part and providing models for calculating the orientation time for spherical particles, fiber, and platelets. An AC or DC field is used to assemble, re-arrange, and orient organic and inorganic graphite and aluminum spherical particles whose electrical properties differ from the properties of the matrix. Dipole-dipole interaction between fibers subjected to an electric field can orient them, leading to the formation of pseudo-fibers [110]. Carbon particles were successfully oriented along the curvature of a circular hole using this technique and the authors have claimed to be able to produce uniform fiber orientations for maximum strength by improving the design of the reported electrodes. Another SLA-like system equipped with electrodes to produce an electric field was used to build parts with rotating anisotropic layers mimicking collagen-fiber alignment in biology and a complex menger sponge structure [116].

Following the footsteps of researchers at the University of Wisconsin, Holmes et al. [85] developed the Field-Aided Laminar Composite or FALCom processing technology which makes use of fillers that align to form pseudo-fibers in a liquid photopolymer solution which can be arranged in a design before being locked in place by laser curing of the photopolymer. These fillers are designed with structural, thermal, and electrical properties for use in multifunctional components for structural, heat sink, or electrical applications. The filler material particles ranged from, 20 nm to 80 μm in size, making it possible to manipulate multiple particles at once in an electric field, to form large, aligned, pseudo-fibers visible to the human eye as seen in Figure 5A. Kim et al.'s research determined that in order to achieve particle orientation in a short amount of time (a few seconds), a matrix viscosity of 1 Pa.s or lower is ideal. Holmes made use of a 0.9 Pa.s photopolymer along with Multi-Wall Carbon Nanotubes (MWCNT), aluminum and alumina microparticles of sizes ranging from 20 μm to 53 μm. A study on the change in viscosity of the matrix as the vol.% of the filler material was raised over 0.5 wt.% and with a change in particle size was conducted. Due to the electrostatic force on the particles, they move towards the electrodes with opposite charges. Also, when two or more of these particles are in an electric field, the dipole-dipole interactions between the particles adjust the relative position of the particles and ultimately align themselves along the direction of the electric field. The aim of their project was to form such pseudo-chains using spherically shaped particles instead of using fibers or rod-shaped particles. As mentioned above, Kim et al. recommend the use of a matrix with a viscosity of 1 Pa.s or lower for faster orientation times based on the study as seen in Figure 5D–F. The response time of particles to the applied field also depends on the material of the reinforcement; carbon fiber was more easily rotated than a glass fiber under the same field and matrix viscosity conditions. The force induced by the field on the filler material creates a torque on the individual particles. This torque is the driving force that rotates the particle to line up with the direction of the electric field [110].

In an advanced and modified version of the same technology, Holmes used a laboratory fabricated stereolithography (SLA) apparatus with electrodes mounted on the crosshead along with the laser source, for electric field generation and to orient particles in the X–Y plane. To achieve through-thickness orientation in the Z plane, an electric field is induced by the build platform. Their goal was to prove the possibility of fabricating 3D structures with 3D reinforcements. However, due to the small particle size, a chain-like or pseudo-fiber reinforcement was achieved instead of a fully continuous reinforcement at high fiber loadings. Also, the reinforcement did not transcend layers to create a bridging effect. Tests conducted with 0.3 wt.% filler resulted in very short and discontinuous chain-like structures. No mechanical tests were performed on the prepared samples to validate improvement in structural behavior [85]. SWNT-polymer composites with controlled electrical and dielectric properties can be produced by actively introducing an AC field [62]. The use of a stereolithography system often raises

issues with suspended filler material as higher weight percentages can lead to sedimentation and high fluid viscosity can impede rapid particle reorientation.

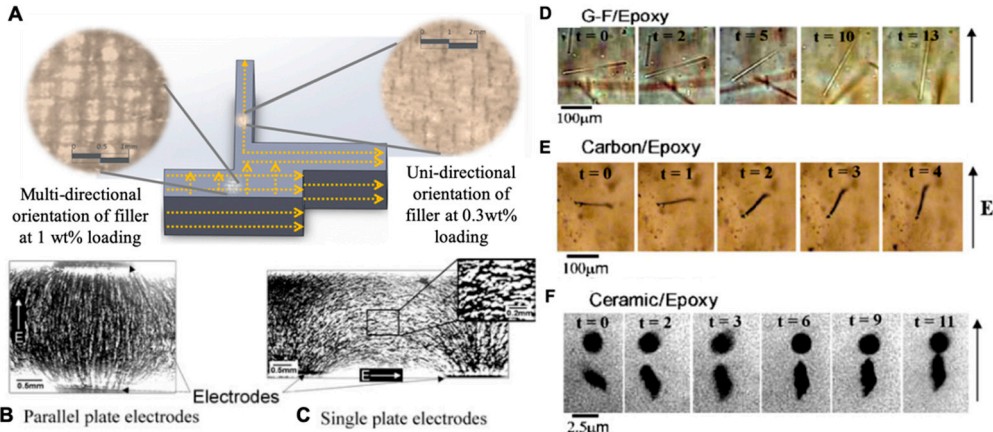

**Figure 5.** (**A**) AM part showcasing directionality and unidirectionality of reinforcement in a single component [85] (**B**) Parallel plate electrodes produce chains oriented through thickness of the material (**C**) Planar electrodes form in-plane chains [117] (**D**) Time sequence and orientation of glass fiber in response to an electric field E = 0.7 kV mm$^{-1}$ (**E**) carbon fiber (**F**) ceramic particles in a liquid epoxy of viscosity 2.5 Pa.s [118].

Electrophoretic deposition or EPD can be classified as a subsection of electric field-assisted AM and has seen some popularity in recent years. Extrusion printing with EPD by Lee et al. [119] implements an electric poling setup between the nozzle and the printed to maintain dipole-alignment in PVDF polymer over a large area. This process is however aimed at obtaining specific polymer chain alignment rather than controlling embedded reinforcement for the creation of β phase crystalline structure. Light-directed electrophoretic deposition involves the deposition of particles in response to light as successfully demonstrated by Pascall et al. on an SLA system [120] enabling the fabrication of complex 3D patterned composites. Traditional composites produced using electrophoresis are limited to producing layers of materials perpendicular to the surface of the electrode, unlike its light-directed counterpart which enables the formation of patterns with different materials in-plane and layering one material over another with the disadvantages being scalability and the need to produce physical masks for each deposited layer. Optically controlled digital electrodeposition (ODE) has shown several advantages over conventional metal patterning processes by allowing dynamic patterning of custom micro-scale silver structures with high conductivities, resolution of 2.7 μm over a large scale in a matter of seconds [121]. The process involves the use of a solution with metal ions injected into an ODE chip followed by projecting images and alternating the electric field to form patterns. The process also enables the control and placement of devices such as microsensors, solar cells and field effect transistors (FET) in addition to nanowires, nanoparticles, and CNTs. Direct ink writing (DIW) system used by Sullivan et al. [122] uses a silver nanoparticle ink with a custom-built cell for producing an electrophoretic system to enable deposition of conductive electrodes with controlled thickness with homogeneous mixing of thermite composites. EPD has been a continuing source of filler control in materials requiring accuracy at a small scale with a wide range of materials [122–125]. An immense amount of research has also been conducted on the orientation of particles and fibers under electric fields either in the absence of a matrix material or using non-AM methods [126–131]. Ounaies et al. explored the use of a combination of both magnetic and electric fields to impart multi-functionality in lightweight, flexible polymer composites. Using origami principles, they produced a material consisting of a dielectric elastomer and a magneto active elastomer to create multi-field responsive structures for application in areas such as reconfigurable aircraft and deployable space structures [35,93,132]. Aligned composites produced using this process displayed tunability of dielectric properties and conductivity by controlling the strength, frequency and time are however produced by casting into molds [129].

Multi-functionality through the embedding of nanomaterials can further extend the capabilities of nanocomposites to properties such as gradients in thermal and electrical conductivity, photonic emissions tunable for wavelength, and increased strength and reduced weight [133–135]. 4D printed structures, soft robots and actuators and other biomimetic shape morphing systems could also take advantage of field-assisted AM for control [13,89–91]. Other promising research has shown improved mechanical properties such as strength, stiffness, fracture toughness, and damping in composites with the inclusion of non-structural functions in the form of electrical and/or thermal conductivity, sensing and actuation, energy harvesting/storage, self-healing capability, electromagnetic interference (EMI) shielding, recyclability and biodegradability, in order to produce a multifunctional composite structure [136–141]. Another important application of polymer composites where electrical conductivity is required is in aircraft structures, where non-conducting structures may be damaged by lightning strikes. Conductive polymer nanocomposites are being investigated as possible replacements for non-conducting polymer matrix materials, thus eliminating the need for add-on metallic conductors, which are too heavy and may be difficult to repair [115,142,143]. Electric field-assisted assembly to control the orientation and placement of solution suspended thin films for electrode manufacturing. Several micrometer-sized monolayer transition metal dichalcogenide (TMD) crystals were positioned on inter-digitated electrodes while almost eliminating defects due to stacking, wrinkling, or folding of sheets. Their results show that the process can be used to additively integrate 2D oriented monolayer materials for nano-devices [144]. Hence, it is clear that field-assisted manufacturing would be a valuable contribution to the field of AM with a well-maintained balance between field and fluid forces.

## 2.3. Acoustic Field-Assisted Additive Manufacturing

The application of an acoustic field for particle manipulation has shown great promise and is said to provide better control over the distribution and orientation of the particles without any material limitations. An excitation frequency in the form of an acoustic field can be used to control fiber orientation in additive manufacturing and unlike magnetic or electric field-assisted manufacturing, the use of an acoustic field does not require the reinforcing material to have a specific property in order for it to respond to the applied field. This technique is primarily used to either concentrate or disperse the reinforcement in certain areas of the print; however, it may be challenging to rotate the particle, as is possible using the methods mentioned above. The distribution of the reinforcement is adjusted by changing the voltage and frequency, thus making it possible to imitate continuous fiber composites, which can typically be difficult to use with additive manufacturing. Yunus et al. [87] made use of a DLP-based SLA printer to apply 100 V at 2.33 Hz and 1400 m/s speed of sound, to align a combination of magnetite nanoparticles, copper nanoparticles, and carbon nanofibers to produce 1D patterns of continuous lines in different layers to test for conductivity of the printed circuitry and tested using an LED. The concentration of fillers determines the thickness of the pattern, meaning that higher filler concentrations would result in better connectivity between layers. They succeeded with a combination of 4 wt.% copper nanoparticles and 100 μm layers, since copper exhibited a much lower resistivity compared to the other materials, thus improving conductivity. A higher pattern height was also attained with the use of varying concentrations of fillers and materials. The height of the pattern was key to enable points of electrical contacts between consecutive printed layers as seen in Figure 6E–J.

Fiber alignment control during direct deposition/extrusion printing has also been demonstrated using piezoelectric actuators within the print head. Collino et al. used an acoustic field-assisted deposition method to align SiC whiskers in an epoxy matrix using microfluidic print nozzles coupled with piezoelectric actuators [86,145]. The application of an excitation frequency caused the SiC fibers to align along the core of the print line whereas the fibers spread apart uniformly when the excitation frequency was turned off. Although this technique allows for fiber control, they were only oriented in the direction of flow and were not seen to provide any additional strength in the Z-direction of the printed structure. Tensile tests comparing the base ink to the aligned print demonstrated improved strength and modulus and decreased ductility in the X–Y plane alone. It is usually expected that

the fibers will align in the print direction without the use of an externally applied field due to shear forces exerted on the particles during flow, shown in Figure 6. Although Collino proved that these fibers could be further controlled using their technique making long continuous chains of focused fibers inside the printed epoxy along the print line, the modulus of the print was compromised. It was hypothesized that the focused samples had a difference in stress transfer from fiber to matrix as suggested by the Halpin-Tsai model. The aggregated fibers have a lower surface-area-to-volume-ratio due to closely arranged fibers leading to lower composite modulus for a given volume fraction as seen in Figure 6B. This process seems promising for modulating the microstructure locally to create functionally graded composites. However, no effort was made to align the particles in such a way to improve the strength in the Z-direction of the print, causing the issue of interlayer bonding to still remain [87]. Ultrasonic fields have also shown similar results of focusing glass fiber reinforcements resulting in the formation of a pattern that matches the ultrasonic field lines similar to patterns produced by an electric field showcasing the versatility of the process, the ability to achieve a wide range of fiber orientations and tunable thermal properties [146–148]. Structures with high and low coefficient of thermal expansion (CTE) provide zero and tunable thermal expansion, lattices with two materials of contrasting linear coefficients give rise to a negative Poisson ratio can be used to improve thermal stability, reduce weight and extend the life of many components and structures [148]. Other composites produced with aligned reinforcement for various applications reported non-AM approaches [149–153]. It is noticeable that most research on field-assisted additive manufacturing processes is solely focused on achieving alignment of reinforcement in the X–Y plane either for aesthetics or to overcome anisotropy. Mechanical properties of these materials in the X–Y plane and in the Z-direction are seldom reported.

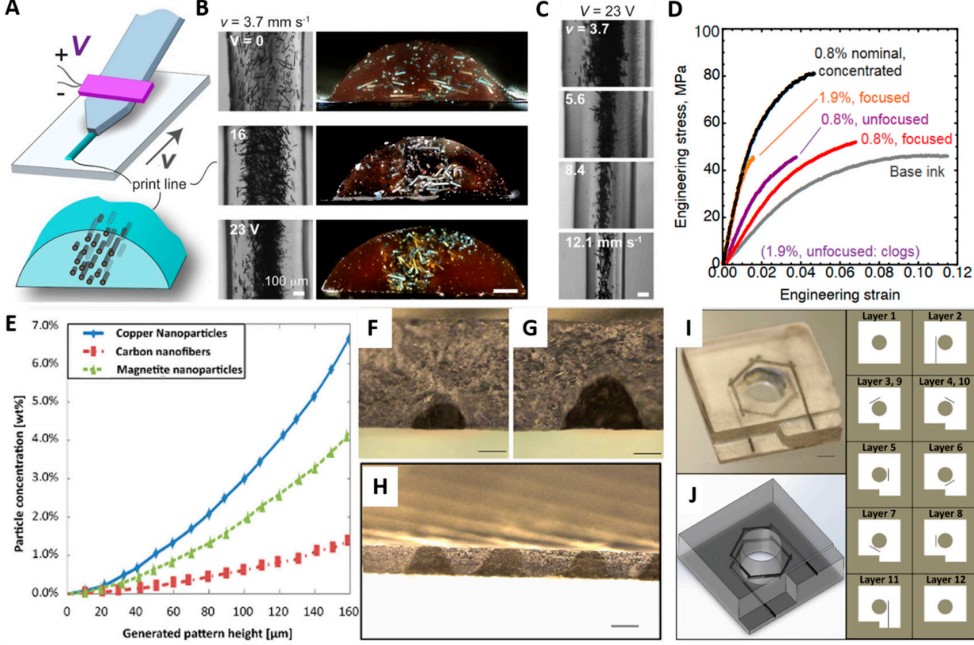

**Figure 6.** (**A**) Schematic of a microfluidic print nozzle with a coupled piezoelectric actuator driven at a peak-to-peak voltage (V), with deposition speed controlled by the substrate velocity (V). (**B**,**C**) Print line microstructures for a constant deposition speed of 3.7 mm s-1 and increasing excitation voltage from 0 to 23 V in plain view and cross-section (**D**) Mechanical characterization of SiC/epoxy inks for tensile tests of the base ink material (epoxy, no fibers) as well as inks with 0.8 vol.% SiC fibers for unfocused and focused condition [86] (**E**) Relationship between reinforcement concentration and pattern height for different materials (**F**) sample cross-section with 50 μm reinforcement height (**G**) 100 μm reinforcement height in a 200 μm sample (**H**) 100 μm reinforcement height in a 100 μm sample (**I**) 12 layer embedded electromagnetic coil pattern surrounding magnetic core cavity (**J**) 3D schematic of electromagnet pattern in UV resin [87].

Ultrasonic additive manufacturing (UAM) can also be classified as an "acoustic field-assisted" form of AM typically used for the production of metal matrix composites at or near room temperature allowing the integration of smart and passive components [154]. Thermally sensitive components can also be integrated into solid metal structures owing to its low processing temperatures unlike in SLS and Selective Laser Melting (SLM) which ultimately imparts residual stresses and introduces undesirable distortion [155]. However, UAM processes do not involve polymer matrices and are hence excluded from this review.

Field-assisted additive manufacturing has shown the capabilities of fine-tuning microstructures of the printed material, either to improve strength or for applications in smart polymer composites for use in soft robotics, biomedical devices, and autonomous systems. As discussed throughout this review, the ability to accurately control filler material with properties for specific applications has shown capabilities such as sensing, self-assembly, and self-healing.

*2.4. Modeling and Orientation Prediction*

Since the orientation of fibers in a composite is one of the most significant factors that determine the overall strength of the final part, an immense about of time and research has been invested in modeling and studying their effects [156–159]. Programs such as MoldEx3D, Abaqus, Fibersim and Digimat provide the capability to predict the strength of an injection-molded composite based on the mold geometry, flow injection ports and streamlines, flowrates, and viscosity, to name a few [160–164]. However, there seems to be a gap concerning modeling for additive manufacturing of short fiber composites and the ability to determine fiber orientation in-flow and post-deposition. Most nozzles used in extrusion-based AM are either convergent or cylindrical with simple geometries that generate shear-driven fiber alignment which can often result in Jeffery orbits, causing the filler particles or fibers to continuously rotate, thus resulting in a random fiber architecture [35,57,165,166]. Furthermore, for field-assisted AM, there is a lack of ability to predict fiber placement and orientation in a final printed part based on the strength of the applied field, reinforcement shape, matrix viscosity, flow parameters, nozzle geometry, and the time it takes for the reinforcement to achieve the desired alignment. As discussed earlier in Sections 2.1 and 2.2, a few articles have investigated the relationship between the fluid forces and externally applied fields for both SLA and extrusion processes, which is still limited in terms of reinforcement material, shape and size [35,84,115,167–169]. As a post-processing step, Herman's orientation factor and other image processing tools in ImageJ and MATLAB have been implemented to determine the alignment of fibers from SEM and microscope images [170–173]. Peridynamic models also have the potential to be adopted from studying damage in complex bio-inspired structures to explore crack propagation and defects in additively manufactured parts [174–176]. Despite the available modeling methods that are currently being modified to be used for AM, there is a need for further development in the aspect of modeling programs and the ability to study material properties by altering the printing parameters, material properties, and composition.

## 3. Challenges and Future Outlook

A common challenge across the board for AM is anisotropic behavior. The inherent nature of this layer-by-layer manufacturing process invariably introduces differences in microstructure at the core of each layer compares to its boundary, thus resulting in poor bonds and different mechanical behavior in transverse and vertical directions of the print. The ability to incorporate and manipulate microstructures in a one-component additive manufacturing system could not only change the way materials are fabricated and used, but also give rise to newer and more versatile properties. This review sampled research over the past decade in the field-assisted alignment of particles in polymeric fluids. These techniques could advance current additive processing and printing, therefore providing novel opportunities for 3D printing to further migrate towards functional part creation.

In summary, the application of external fields to control microstructure in situ has given researchers the ability to create multifunctional parts using a single material. The shape of reinforcing material

has also shown to play a major role in its functional properties. 1D fibers can strengthen a composite much more than spherical particles assembled in the form of pseudo-fibers. The difference between the tensile strength and modulus in the aforementioned processes differ primarily due to the material composition. However, comparing the strength based on different fiber orientations, a significant amount of improvement is observed and shown in Figure 7 [53,58,59,74,86,110,116]. It can be observed that of the few articles that did report on mechanical properties of the printed composite part, most orientations are either along the 0- or 90-degree alignment. Also, composites printed with fiber reinforcement show higher tensile strengths compared to "pseudo-fibers" formed using spherical particles. Composites printed with platelets show the lowest mechanical properties regardless of their alignment and the type of field applied showed no significant variation in the degree of alignment or the duration to achieve alignment.

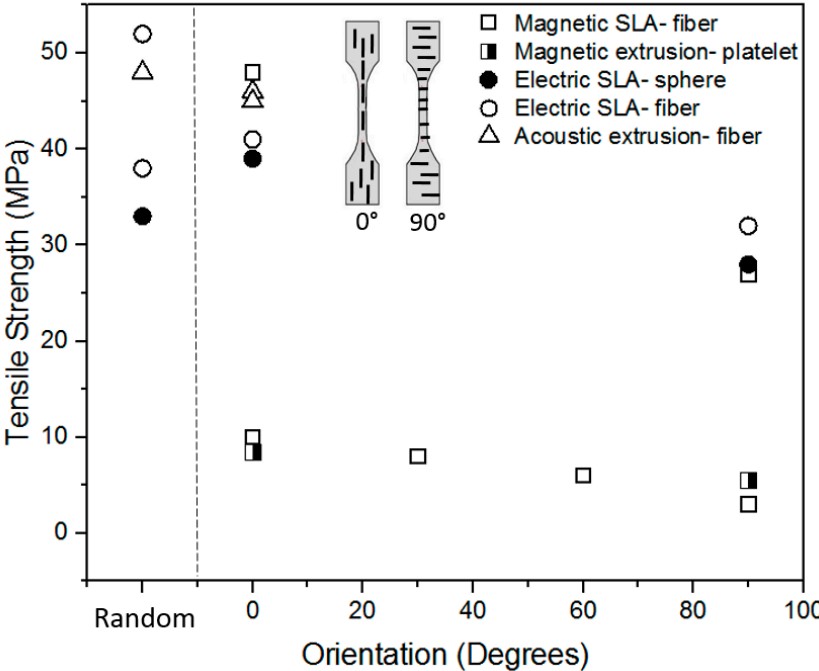

**Figure 7.** Tensile strength of various composites manufacturing using SLA or extrusion AM with applied magnetic, electric, and acoustic fields [53,58,59,71,82,107,113].

For magnetic field-assisted SLA processes, the mechanical properties improve from 3–10 MPa as the orientation of the fillers range from 90 to 0 degrees respectively, while orienting fillers in an osteon or concentric circular pattern exhibits a 9 MPa strength in all directions. Electric field assisted SLA shows a comparison between the use of spheroids and fibers as fillers, proving that fibers act as a superior reinforcing filler. However, most field-assisted stereolithography references primarily focus on proof of concept or achieving functionality such as conductivity and actuation using fillers responsive to external fields, rather than improving mechanical properties. A limited study has been reported on the polymer-filler interface for composite used in AM, which is key in preventing fiber pullout and delamination by creating strong chemical bonds between compatible materials [110–114,133]. For extrusion-based AM, it can be noted that fiber or particle agglomeration along with high volume percent can result in the nozzle being clogged. It is crucial to use particles in a size that is an order of magnitude smaller than the nozzle opening to avoid blockage issues and interruptions in prints. Higher viscosity composites require higher external fields as well as longer alignment times, which is difficult to achieve in extrusion processes due to a continuous material flow unlike in vat photopolymerization. Table 1 below summarizes the type of assisted field used with AM and information on the reinforcing material.

**Table 1.** Materials and processes used in field-assisted AM.

| AM Process | Specific Approach | Field | Filler Form | Concentration | Alignment |
|---|---|---|---|---|---|
| SLA/Vat photo polymerization | Ref. [58] | Magnetic | Ferromagnetic short fibers | 0.5 vol.% | Unspecified |
| | Ref. [59] | Magnetic | Alumina particles/platelets coated with iron oxide nanoparticle, $Fe_3O_4$ | 10, 15 vol.% | Unspecified |
| | Ref. [110] | Electric | Ceramic micro spheres; pseudo-fibers | 5 vol.% | In-plane, 0, 90, and random |
| | Ref. [110] | Electric | Glass fiber | 5 vol.% | In-plane, 0, 90, and random |
| | Ref. [115] | Electric | Multi-walled CNT | 0.1 vol.% | In-plane |
| | Ref. [115] | Electric | Aluminum microparticles; pseudo-fibers | 1 vol.% | In-plane |
| | Ref. [87] | Acoustic | Magnetite and copper nanoparticles, carbon nanofibers | 0–9 wt.% | In-plane, 0, and 90 |
| Extrusion | Ref. [35] | Magnetic | Calcium Phosphate rods with $Fe_3O_4$ NP coating | 0.2 vol.% | In-plane; Perpendicular to print direction |
| | Ref. [53] | Magnetic | Magnetized anisotropic stiff platelets, non-magnetic (alumina platelets) particles with iron oxide NP | Tunable concentration | In-plane; custom alignment due to rotating field |
| | Ref. [86] | Acoustic | SiC fibers, solid $BaTiO_3$ spheres, hollow $SiO_2$ spheres | 0.8 vol.% | Focused in-plane |
| | Ref. [146] | Acoustic | Glass microfibers | Unspecified | In-plane, 0, 45, and 90 |

## 4. Conclusions

Field-assisted additive manufacturing has shown great potential with regards to microstructure tunability in additively manufactured composites. At the current stage, a wide range of materials and field-assisted manufacturing processes have been explored; however, there remains the need for newer microstructure design and optimization for improved performance of composites, along with an in-depth study of material properties ideal for each additive manufacturing process. Both vat polymerization and extrusion-based printing have shown to successfully incorporate magnetic, electric, and acoustic fields for the manipulation of reinforcement. Although it allows for tailor-made properties, it also means that a lot of factors can affect the final outcome of the print. For additively manufactured composites to possess the same level of mechanical strength as bulk material while having multifunctional properties, a thorough understanding of interfacial bonds and fiber-matrix and between layers is necessary, which will help bridge the gap between parts produced using these methods.

**Funding:** The authors gratefully acknowledge the generous support from the Center for Complex Materials Design for Multidimensional Additive Processing (CoManD), NSF award #1735968 and Research Infrastructure for Science and Engineering (RISE), NSF award# 1646897. We would also like to thank the FAMU-FSU College of Engineering and the High-Performance Materials Institute.

**Conflicts of Interest:** The authors declare no conflict of interest.

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
