# Peer review of "Composite Reinforcement Architectures: A Review of Field-Assisted Additive Manufacturing for Polymers"

_jcs, doi:10.3390/jcs4010001_

Round 1

Reviewer 1 Report

This review is well-written and comprehensively described the significance of field-assisted aligning of nanostructured materials during additive manufacturing. One minor suggestion I have is to encourage the authors to add some relevant papers that reports field-assisted aligning during manufacturing, 1)ACS applied materials & interfaces, 2016 8 (28), 18471-18480; 2) Nanoscale, 2015, 7 (35), 14636-14642 and 3) ACS applied materials & interfaces, 2015, 7 (7), 4306-4310

Author Response

Thank you for your review. The suggested references have been included in the document as “In addition, such composites have also seen applications in electronics for their controllable conductivity, capacitance and optical properties with an advantage of roll-to-roll on-demand production capabilities [63–65]” in section 1.1.

Reviewer 2 Report

In this article, the authors review numerous investigations
about the ability to control
composites microstructure by applying various types of external
field during manufacturing and processing.
In particular, the applications of magnetic,
electric and acoustic fields are extensively discussed.
This paper should be accepted for publication in its present form.
It is a valuable contribution
for investigators working in this area.

Author Response

Thank you for your review.

Reviewer 3 Report

This review manuscript entitled ‘Composite Reinforcement Architectures: A Review of Field-Assisted Additive Manufacturing for Polymers’ well described historical and fundamental backgrounds of the field and as well as its perspective. This manuscript will be a useful resource to a broad range of researchers and I recommend this manuscript for Journal of Composites Science, as is.

Author Response

Thank you for your review.